# A Spectrum of Age- and Gender-Dependent Lower Urinary Tract Phenotypes in Three Mouse Models of Type 2 Diabetes

**DOI:** 10.3390/metabo13060710

**Published:** 2023-05-31

**Authors:** Bryce MacIver, Erica M. Bien, Mariana G. de Oliveira, Warren G. Hill

**Affiliations:** 1Laboratory of Voiding Dysfunction, Nephrology Division, Department of Medicine, Beth Israel Deaconess Medical Center and Harvard Medical School, 99 Brookline Ave., Boston, MA 02215, USA; imaciver@bidmc.harvard.edu (B.M.); ericambien@gmail.com (E.M.B.); 2Department of Pharmacology, Faculty of Medical Sciences, University of Campinas (UNICAMP), Campinas 13083-970, SP, Brazil; gdeoliveira.mariana@gmail.com

**Keywords:** diabetic cystopathy, diabetic bladder dysfunction, micturition, urology, insulin resistance, incontinence

## Abstract

Lower urinary tract symptoms are extremely common in people with diabetes and obesity, but the causes are unclear. Furthermore, it has proven difficult to reliably demonstrate bladder dysfunction in diabetic mouse models, thus limiting the ability to gain mechanistic insights. Therefore, the main objective of this experimental study was to characterize diabetic bladder dysfunction in three promising polygenic mouse models of type 2 diabetes. We performed periodic assessments of glucose tolerance and micturition (void spot assay) for eight to twelve months. Males and females and high-fat diets were tested. NONcNZO10/LtJ mice did not develop bladder dysfunction over twelve months. TALLYHO/JngJ males were severely hyperglycemic from two months of age (fasted blood glucose ~550 mg/dL), while females were moderately so. Although males exhibited polyuria, neither they nor the females exhibited bladder dysfunction over nine months. KK.Cg-A^y^/J males and females were extremely glucose intolerant. Males exhibited polyuria, a significant increase in voiding frequency at four months (compensation), followed by a rapid drop in voiding frequency by six months (decompensation) which was accompanied by a dramatic increase in urine leakage, indicating loss of outlet control. At eight months, male bladders were dilated. Females also developed polyuria but compensated with larger voids. We conclude KK.Cg-A^y^/J male mice recapitulate key symptoms noted in patients and are the best model of the three to study diabetic bladder dysfunction.

## 1. Introduction

Urologic complications of diabetes are among the most common benign urological diseases affecting both men and women across all ages in the United States [1]. Collectively, it is estimated that these complications, including a spectrum of urinary incontinence (UI) and lower urinary tract symptoms (LUTS), could affect up to 87% of patients diagnosed with diabetes mellitus (DM) [2,3]. Extensive literature demonstrates that women with obesity/type 2 diabetes (T2D) are at a much higher risk of developing UI [4,5,6,7,8,9,10,11,12]. Although few studies have specifically distinguished between types of UI (mixed, urge, or stress), evidence suggests that obesity/T2D are significant risk factors for all three [4,13,14]. It has long been known that LUTS is highly prevalent in aging men, with estimates ranging from 25% in men 40–49 years old to 80% in men 70–79 years old. Recent reports provide strong support for an association between obesity/T2D and benign prostatic hyperplasia (BPH) and LUTS [15,16,17,18] in men.

While the association between diabetes and LUTS is strong, the natural history of these conditions is highly variable, and the etiology is almost completely unknown. The development of symptoms occurs over decades and has been attributed to neuronal, smooth muscle, and urothelial dysfunction, among others [19,20]. Given the complexity and difficulty of studying the pathophysiology in human patients, it is logical to turn to animal models, which offer the advantages of direct access to tissue, shorter life spans, and in the case of mice, the ability to test hypotheses through genetic manipulation.

Studies in mice and rats have been very useful, particularly for studying diabetic voiding dysfunction (DVD) in type 1 diabetes (T1D), due to the relative ease of using high-dose streptozotocin [21,22,23] or via monogenic mouse models such as the Akita strain which have a mutation in the insulin 2 gene that results in endoplasmic reticulum (ER) stress and toxicity to pancreatic β cells [24,25]. Likewise, the effects of obesity on voiding dysfunction have been investigated using high-fat diet models or through the use of monogenic leptin-deficient models (Lep*^ob^* and Lep*^db^* mice) [26,27]. Many studies, however, use relatively young mice or short time frames, e.g., 8 to 16 weeks, for interventions such as streptozotocin or high-fat feeding and draw conclusions based on endpoints which are often chosen for cost and/or convenience. In recent years, however, an appreciation has developed for the changing nature of voiding dysfunction in animal models when studied over longer time frames [19,28,29]. This has resulted in a ‘temporal hypothesis’ that generally posits compensation and decompensation stages in bladder function that manifest as early-stage overactivity and later-stage underactivity, respectively. In this regard, rodent models appear to share some of the same pathological sequelae that are seen in human patients but that in humans develop over decades.

Type II diabetes (T2D) has been a more challenging problem in terms of creating animal models because it is a heterogeneous disorder that requires both genetic and environmental factors to interact. At its simplest, it can be described as a pathology characterized by hyperglycemia deriving from either absolute or relative insulin deficiency. Obesity is the most common etiological factor in T2D but is not an absolute requirement, with the key pathophysiological features being impaired insulin secretion and/or decreased insulin sensitivity (insulin resistance) [30]. In efforts to create better and more representative mouse models for T2D, there has been a move toward crossing strains that combine several characteristics of the disease in ways that avoid some of the extreme and rapid manifestations of monogenic models, such as the Akita and Lep*^ob^* mice. By creating polygenic strains that exhibit variable onset times and reduced severity of features such as obesity, hyperglycemia, and insulin resistance, the hope is to more closely model the clinical features of at least some populations within the human T2D universe.

We believe that finding a mouse model would be very useful in understanding the intrinsic mechanisms of diabetic bladder dysfunction. Therefore, the main objective of this experimental study was to characterize diabetic bladder dysfunction in three polygenic strains of T2D. The primary endpoint was to identify which strains would most closely resemble the human disease in terms of an evolving phenotype, and the second endpoint was to define the timeline and dietary regimen required for symptom onset.

## 2. Materials and Methods

To achieve our goals, we used three polygenic strains of T2D, subjected to regular glucose tolerance and VSA tests for 8 to 12 months. The three strains, KK.Cg-Ay/J (KK-Ay), TALLYHO/JngJ (TallyHo), and NONcNZO10/LtJ (NoncNZO) were selected to represent different aetiologies of T2D. Both males and females were evaluated, as well as the effect of normal and high-fat diets. These three strains were compared with each other and with normoglycemic C57BL/6J mice, the data of which was obtained from literature reports and public databases.

All experiments on mice were performed with the approval of the BIDMC Institutional Animal Use and Care Committee under approved protocol #014-2021.

### 2.1. Study Design

Mice of three type 2 diabetic strains were obtained between the ages of four and seven weeks from The Jackson Laboratory (Bar Harbor, ME). In total, there were 20 KK-Ay, 20 TallyHo, and 24 NoncNZO, half male and half female. Females and males were further divided in two, and half placed on a ‘control’ diet (16% calories as fat) and half on a moderately high-fat diet (26% of calories provided as fat), at either 4 weeks of age (KK-Ay and TallyHo) or 8 weeks of age (NoncNZO—the earliest age at which this strain was available). All mice were placed on filter papers to perform voiding spot assays (VSA) at baseline (either 4 or 8 weeks of age). Then, at monthly intervals, they were weighed, VSA tested, and glucose tolerance testing (GTT) was performed.

We have found, over a number of years of characterizing VSA in mice of different strains and in different models, that several key parameters can be used to describe voiding phenotypes [29,31,32]. Total urine volume is useful for assessing polyuria. Primary voids (PVs) refer to the number of (what we consider) true voiding events and are represented by voids larger than 20 μL (the justification for this cutoff may be found in [33] and is based on frequency distributions of C57BL/6J spot areas from 220 filter papers). Primary void spot (PVS) volume is the average volume of the PVs in a four-hour period. Finally, microvoids are the number of spots from 2–20 μL and may indicate incontinence, outlet obstruction, or bladder pain if they are numerous.

### 2.2. Diets

Irradiated diets were provided by LabDiet (St. Louis, MO, USA). The regular fat diet was JL Rat & Mouse 6F Irrad (5LG4) and is similar in ingredients and macronutrients to the 5K52, which is the primary feed formula used by The Jackson Laboratory and contains 16% of calories as fat. The higher fat diet, which was provided to more closely mimic a Western diet with additional fat content, was the Picolab High Energy Mouse Diet (5LJ5), which contains 26% of calories as fat.

### 2.3. Mouse Strains

All mice were sourced from The Jackson Laboratory (Bar Harbor, ME, USA).

*TALLYHO/JnJ* (referred to as TallyHo or THo mice)—Strain #005314: 12 male and 12 female 4-week-old mice were used. Males and females were divided into two groups each, half were provided a diet containing 16% fat by calories, and half were given a moderately high-fat diet containing 26% fat by calories (referred to as ‘the high-fat diet or HFD’).

*KK.Cg-A^y^/J* (referred to as KK-Ay mice)—Strain #002468: 12 male and 12 female 4-week-old mice were used. Males and females were divided into two diet groups, as for the TallyHo mice.

*NONcNZO10/LtJ* (referred to as NONc mice)—Strain #004456: 20 male and 20 female, obtained at 7–8 weeks of age (earliest available age from JAX). Males and females were divided into two diet groups, as for the TallyHo mice.

### 2.4. Strain Phenotypes

Detailed descriptions of strain development, genetics, and phenotypes can be found at www.jax.org (accessed on 1 December 2021), but all are considered polygenic models of type 2 diabetes (T2D), having been derived by cross-breeding different strains to achieve particular traits associated with obesity, hyperglycemia, and insulin intolerance (see Table 1 for an overview of these characteristics).

### 2.5. Void Spot Assay (VSA)

VSAs were performed in the afternoon, as described previously [29,32]. Mice were moved individually to empty mouse cages with precut filter paper (Blicks Cosmos blotting paper #10422–1005) covering the bottom. They were provided with food in their usual wire racks but no water. After 4h, mice were returned to their home cages, and the filter papers were allowed to dry before being photographed under UV light (365 nm) in a Chromato-Vue C75 imaging box with an onboard Canon camera (EOS Rebel T3–12 megapixels). Before image analysis, any overlapping spots were outlined with the drawing tool in the Fiji version of Image J, copied, and moved to an empty area of the filter. Void spot areas were quantitated using an in-house-developed machine learning object recognition algorithm. The results were compared with our formerly published UrineQuant software [29], and the results for all parameters demonstrated correlation coefficients of R^2^ > 0.98, thus validating its accuracy and consistency with previously published data from our group [29,32,33,35].

### 2.6. Glucose Tolerance Testing (GTT) 

Mice were fasted for 5 h on the morning of testing before blood glucose was measured with a One Touch Ultra2 glucose meter and test strips on samples obtained through tail nick bleeding (~2 mL). Time zero fasted blood glucose was taken before the intraperitoneal (i.p.) injection of glucose (2 g/kg), followed by repeat testing at timed intervals (15, 30, 60, 90, 120 min) after glucose administration [29,36]. Note that the glucometer used for this study has a maximum readout of 600 mg/dL, so the true maxima for TallyHo males are unknown.

### 2.7. Histology

Masson’s Trichrome staining on paraffin-embedded formalin-fixed cross-sectioned bladders (5 mm sections) was performed by the BIDMC histology core using standard methods [37]. Fixation was achieved in 4% paraformaldehyde:PBS (phosphate buffered saline) prepared from 16% stock solution (Cat# 15710, Electron Microscopy Sciences, Hatfield, PA, USA) by immersing whole bladders in 0.5 mL overnight at 4 °C. Bladders were then washed three times in PBS. Slides were imaged using bright field microscopy on an Olympus BX60 microscope using 4× and 10× objectives, and images were acquired on a computer using CellSense software.

### 2.8. Statistics

All data except AUC (Area Under Curve) are expressed and shown graphically as means ± standard deviation (SD). AUC was calculated using Graphpad Prism 9 software with 100 mg/dL set as the baseline for analysis. AUC data are shown with standard error because Prism provides the results of the analysis in this manner. Statistical analysis was performed in Graphpad Prism using two-way ANOVA with post-hoc testing for multiple comparisons (either Sidak’s or Dunnett’s multiple comparisons tests). For comparisons of bladder:body weight ratios between males and females, t-test with Welch’s correction (no assumption of equal SDs) was used. Statistical significance was defined as *p* < 0.05.

## 3. Results

### 3.1. Body and Bladder Weights

Figure 1 shows weight changes with age and reveals that the three strains showed gender and strain differences, but the high-fat diet had only a relatively modest, or no effect, on weight gain compared to the control diet. Interestingly, the females of two strains showed greater weight gain than the males. In general, the NoncNZO mice exhibited no tendency toward obesity, whereas females of the other two strains became obese. KK-Ay males gained weight moderately (~40 g at 8 months), while TallyHo males actually declined in weight after 2 months. This may be related to their extreme degree of hyperglycemia from a young age (discussed in Section 3.2 Glucose tolerance testing). It can be seen in Figure 1 that mice were weighed for a varying number of months. This was not arbitrary but related to survival. This study was initially designed to run for 12 months, and for NoncNZO mice, that was achieved. However, KK-Ay males began to die from the age of 7 months, and the study was terminated for that strain at 9 months as we needed tissue for histology. The TallyHo males also began to exhibit mortality, so phenotyping was ended for this strain at nine months. Death ratios are shown in Table 2, and while the numbers are small, there was clearly a higher mortality rate for mice on high-fat diets.

Of particular interest from the perspective of bladder phenotypes is the ratio of bladder weight to body weight. As mice become heavier and larger with age, the bladder has a natural tendency to become bigger as well, and in general, the ratio remains fairly constant with aging, at least for C57BL/6J mice [38]. Figure 2A shows the bladder:body weight ratios for all strains (mg:g), and these data revealed some dramatic differences in bladder sizes and ratios. KK-Ay males had enormous bladders which weighed around 200 mg (compared to ~30 mg for 30 g C57BL/6J mice, with a ratio of 1.0 indicated by the dotted line [38]). When the KK-Ay males were sacrificed at 8 months, many had full and clearly enlarged bladders, as shown in Figure 2B (dashed line showing the bladder outline). In this example, the diameter of the bladder along the longest axis was a remarkable 1.9 cm. TallyHo males also had large bladders for their body weight, but in striking contrast, the females of this strain had very small bladders (Figure 2A). Masson’s trichrome staining on mid-bladder sections is shown in Figure 2C,D. The images taken under 4× objective are presented so that the relative bladder sizes are all comparable. The KK-Ay males had massively dilated bladders with thin walls (Figure 2C, left), while the females were unremarkable (Figure 2D, left). The TallyHo males were very large (in this example, extending some distance to both the left and right beyond the frame boundary, Figure 2C, middle) but morphologically relatively normal in having a thick detrusor and relatively well-organized lamina propria (blue) and urothelium. The females, in contrast, had small bladders (~20 mg) for their body size (~50 g) (Figure 2D, middle). The NoncNZOs were unremarkable.

### 3.2. Glucose Tolerance Testing

Examples of glucose curves are shown at different ages for all strains (Figure 3, Figure 4 and Figure 5) to illustrate changes in glucose tolerance over time. However, a more useful way of visualizing the dynamic response of males and females to glucose, and the effects of different diets, is by ‘area under the curve’ or ‘AUC’, which provides a single value for each group at each time point.

Figure 3 shows the results from KK-Ay mice. By three months, both males and females become extremely glucose intolerant (Figure 3B) and significantly worse than at one month (Figure 3A). Baseline fasted glucose values were 224 and 299 mg/dL for females and males, respectively, at 1 month and 374 to 456 mg/dL at 3 months. By way of comparison, five-month-old C57BL/6J male mice have six-hour fasted blood glucose values of ~150 mg/dL [36]. Based on the GTT curves at 6 and 8 months (Figure 3C,D) there is some modest improvement for all KK-Ay groups. Figure 3E shows AUC values for all groups at different ages. Again, by way of comparison, C57BL/6J males exhibit AUCs of ~12,500 (indicated by dashed line) [36], thus emphasizing just how glucose intolerant the males and females of this strain are. There is some improvement shown for all but the females on HFD by eight months.

The GTT results for TallyHo mice are shown in Figure 4. At one month, they do not appear to be too metabolically abnormal (Figure 4A), although they are still modestly hyperglycemic at baseline (~200 mg/dL). By three months, however, the males have extremely high fasted glucose levels of between 500 and 600 mg/dL. After three months, we chose not to inject a glucose bolus for further GTT testing in these males (to avoid the potential for hyperglycemic shock) but simply measured five-hour fasted glucose levels, which were always consistently above 500 mg/dL and often >600 mg/dL. In Figure 4C, the dotted lines indicate the fasted baseline values extrapolated for the male groups.

Figure 4D shows the TallyHo AUCs. The males were maximal at 2 months with values of 56–60,000 and were the most severely hyperglycemic of any strain or gender. For females, there was clearly a diet-related effect, with the HFD negatively impacting glucose tolerance. For females on the control diet, the glucose intolerance was modest and did not change much with aging.

NoncNZO mice were the least hyperglycemic and most glucose tolerant (Figure 5) of the three strains. Males were slightly worse than females. The HFD had a modest deleterious effect on both males and females (Figure 5D). Females on a regular diet were essentially indistinguishable from healthy C57BL/6J mice (dashed line).

### 3.3. Void Spot Assays

When all of the data were collated and statistically analyzed, it was found that there were no significant diet-dependent differences in the overall voiding phenotypes of each strain. As a result, the data for mice on control and high-fat diets have been combined to make the presentation of trends clearer. Figure 6 shows the results from testing KK-Ay mice on filter paper every month for eight months. Each of the four parameters is shown comparing males to females; however, one of the main aims of the study was to define changes in voiding behavior over time since T2D is a chronic illness for which symptoms in human patients worsen with time. Therefore, the statistical comparisons shown are between one-month data and subsequent time points. Figure 6A shows that both males and females developed polyuria, with total urine volumes peaking at 4–5 months and then diminishing somewhat.

Figure 6B reveals that female KK-Ay mice produced a consistent number of primary voids over eight months (four to five); however, Figure 6C shows their average volume/void increased dramatically, peaking at five months. Meanwhile, the males did the opposite, with a significant increase in the number of voids at four months (Figure 6B) which then diminished back to baseline, while the average volume/void (Figure 6C) did not vary at any age. Therefore, if we focus on micturition at four months for males, they urinated more often (Figure 6B) to accommodate the polyuria (Figure 6A), but the volume urinated did not increase (Figure 6C). The male mice were also characterized by an ever-increasing number of microvoids, and these changes were highly significant (Figure 6D). Representative filter images (from five months of age) of female and male voiding patterns are shown in Figure 6E,F, respectively. The female has few but very large PVs, while the male has many spots, some of which would qualify as primary voids and many as microvoids. Both are clearly polyuric, but the phenotypes are completely different.

An interesting additional comparison to emphasize the different adaptive responses of males and females can be seen in Figure 7. Here, we superimposed data for fasted blood glucose (dashed green line) with total volume (yellow line) and PVS volume (red line) in KK-Ay mice. The females in Figure 7A show a striking concordance in the curves, which clearly illustrates an association between blood glucose, polyuria, and volume/void. The total urine produced by males (Figure 7B) also recapitulates quite closely the rise and fall in fasted glucose values; however, despite having to urinate a much greater volume in four hours, there was no change in volume/void. If we attempt to integrate these divergent phenotypes with the bladder:body weight ratios and the histology shown in Figure 2, it would appear that females were able to accommodate the increased urine flow from the kidney, i.e., the bladders filled up more, but they voided at the same frequency (~once/hour). The males, in contrast, exhibited a classic biphasic diabetic response in which the bladder appeared to compensate by voiding more frequently at 4–5 months (overactivity?), but by 6–8 months, it had decompensated (Figure 6B). At this stage, they became enlarged and voided less frequently while simultaneously exhibiting a loss of outlet control (Figure 6D). In essence, the bladders became underactive, and the mice became incontinent as they dealt with the polyuria (Figure 6A) by leaking.

Figure 8 shows the same data analysis for TallyHo mice. The TallyHo males were extremely hyperglycemic from 2–3 months of age (Figure 4), and this was reflected in bladder phenotype as pronounced polyuria (Figure 8A,F). The average voided volume from three months on was ~1300 μL or 325 μL/h. There were no obvious deteriorating or changing trends in the male voiding phenotype since the polyuria appeared to be handled consistently throughout the nine months of the study by an elevated number of primary voids (7–8) and slightly larger than normal PVS volume (150–200 μL). The term normal here is used in comparison to C57BL/6J mice which, in a four-hour VSA, normally void 3–4 times in PV volumes of 100–150 μL [33]. Male microvoid numbers were in the range of 10–18, which is fairly typical for male mice, which tend to spot a bit more than females. Female TallyHo mice were consistently within what we have come to accept as the normal range, and the bladders showed no signs of dysfunction out to nine months. Figure 8E shows a classic female voiding pattern.

NoncNZO mice were the most normal in both male and female voiding phenotypes (Figure 9). In general, the phenotypes were fairly stable over 12 months, although female total urine volume slowly increased. There were relatively minor differences between males and females. Both voided about the same total volume, but the males tended to urinate slightly more often (PVs) in smaller volumes. This can be seen in representative filter patterns shown in Figure 9E (female) and Figure 9F (male). While a couple of age points achieved transient significance, there was no consistent trend toward a deteriorating micturition phenotype. 

## 4. Discussion

Good mouse models of diabetic bladder dysfunction (DBD) for type 2 diabetes are lacking. Therefore, we performed a longitudinal study on males and females of three polygenic strains that, heretofore, have not been investigated for the time-dependent emergence of voiding phenotypes.

DBD is among the most common and incapacitating complications of diabetes mellitus, causing urinary incontinence and poor emptying of the bladder [39]. Despite its prevalence, its natural history is unclear. For many people, an early sign of diabetes is excessive production of urine (polyuria), arising from the inability of the proximal tubule to reabsorb the elevated sugar levels filtered by glomeruli in the kidney. The increased osmolality of tubular fluid produces an osmotic diuresis with elevated urine output. As a result, urinary incontinence in patients was historically attributed to overflow incontinence, i.e., overfull bladders leaking in the absence of any urge to urinate [40]. This view has been challenged, however, by other studies presenting evidence of increased frequency, urgency, and urge incontinence in diabetes [41,42,43]. Polyuria was convincingly demonstrated in several of the mouse models studied here. Ultimately however, diabetic cystopathy is generally considered to occur in later stages (after years or decades of poor glycemic control) and be characterized by reduced bladder sensation, increased capacity, impaired emptying, and elevated post-void residual volume [39,40,43]. 

Clearly, one of the challenges with attempting to model such heterogeneous phenomena in rodents is that it is impossible to know what sensations the animal is experiencing. While there are invasive techniques, such as cystometry, that can quantitate bladder compliance, peak pressure, and residual volume to some degree, they can only be employed once and are terminal. The advantage of the VSA is its ability to track changes in lower urinary tract readouts over time since it does not harm the animal. Its disadvantage is that these readouts are limited to volumes, voiding frequency, and spotting as a proxy for incontinence. Despite this, the results presented here clearly point to some strains and genders as better models than others for future mechanistic studies aimed at understanding the etiology of DBD.

It is clear from the spectrum of evolving phenotypes we observed, both metabolically and in terms of voiding physiology, that it is not a simple matter to select a diabetic bladder dysfunction model that will satisfy the needs of all investigators. Considerations of gender, diet, strain, and timing are all important, and choosing models will depend on the questions being asked and the hypotheses being tested. This study, at a minimum, provides an overview of the effects of chronic hyperglycemia (of varying severity) on voiding physiology and the changing landscape with age in three models of T2D. The mortality results also provide an estimate of how long a study may run, as well as the number of animals required to meet scientific objectives. 

### 4.1. Broad-Based Themes to Emerge

(1)Hyperglycemia usually drives polyuria, but not always. The positive correlation applied to KK-Ay males and females and to TallyHo males but did not apply to TallyHo females. NoncNZO females were not hyperglycemic and did not exhibit polyuria, while the males were modestly hyperglycemic and also were not polyuric. In the case of KK-Ay mice, the correlation between fasted blood glucose (which constantly changed over the lifespan of the study—8 mo) and total urine volumes was extremely tight (Figure 7). Tentatively, it might be concluded that, in mice, the degree of hyperglycemia is critical in driving a polyuric response and that exceeding some glucose threshold triggers it. The TallyHo females, for example, had AUC values of ~40,000 (compared with C57BL6/J’s at 12,500; Figure 4D) but were in the normal range for four-hour urine volume (~400 μL; Figure 8A).(2)The moderately high-fat diet used, 26% calories by fat compared to 16% for control, had minimal effect on weight gain and modest but consistent effects on glucose intolerance (making it worse); however, differences in voiding phenotypes were not detected with VSA. This may be because the VSA is a relatively noisy assay requiring larger group sizes to tease out phenotypes. Careful and appropriately powered cystometric studies may be required to uncover diet-related differences using these diets.(3)The metabolic status of T2D strains can be a moving target that, in some cases, improves with age. We note that KK-Ay males and females, as well as TallyHo females, deteriorated in glucose tolerance initially, while at later age points, improved somewhat.(4)The NoncNZO strain does not appear to be a useful model for the study of diabetic voiding dysfunction. The females were normoglycemic for 12 months, while the males were modestly hyperglycemic and stable. Void spot parameters for both males and females were relatively unchanging for 12 months and mostly within, or not too different, from what we would consider a normal range based on C57BL6/J mice for comparison.

### 4.2. Interpretation of VSA Parameters

As noted in the Materials and Methods (Section 2.1), we have developed, over a number of years, a relatively strong justification for dividing urine spots into two size categories. Urine spots with a volume greater than 20 μL represent ‘true’ voids, i.e., deliberate micturition of stored urine. We term these primary voids or PVs. Urine spots smaller than this, i.e., those from 2–20 μL are, in normal healthy mice, present in variable numbers—usually more are deposited by males, perhaps due to marking or territorial behaviors—but regardless of the number, these microvoids (in healthy mice), usually constitute less than 5% of the total volume on the filter paper. Therefore, the periodic and necessary function of bladder emptying, i.e., voiding, is seen in the PVs.

We present four parameters that experience has shown us tend to capture important phenotypic characteristics. These are total urine volume/4 h, number of PVs, mean volume of PVs, and number of microvoids. These are necessarily all related and interdependent. If, for example, two groups of mice being compared have approximately the same total urine volume, but one group has more PVs, then necessarily the volume/void or PV volume in that group must be less. Consistent changes in one direction or the other can point to important phenotypic changes. We see this playing out in interesting and provocative ways with the TallyHo and KK-Ay strains.

### 4.3. TallyHo Mice

For the TallyHo strain, we note that the males were extremely hyperglycemic and glucose intolerant by two months, while the females were less so but still well above control levels (Figure 4). Analysis of voiding patterns showed that the males became persistently polyuric from two months, with an average four-hour voiding volume of 1300 μL, i.e., 325 μL/h, while the females exhibited no sign of elevated void volumes (consistently voiding 100 μL/h), despite being in a fairly substantial hyperglycemic state (Figure 8). With the exception of the initial jump in micturition volume for males, none of the parameters for either males or females changed much over nine months. There was no evidence for overactivity (more primary voids, smaller PVS volume), underactivity (fewer primary voids, larger PVS volumes, possibly more microvoid activity), or incontinence (elevations in microvoids). Therefore, phenotypically, this strain shows no DVD by females at all, and for the males, which are a tremendous model of profound hyperglycemia, they may be useful as a model of diabetic polyuria but not of DVD.

### 4.4. KK-Ay Mice

Both males and females of the KK-Ay strain were extremely glucose intolerant from three to six months, after which there was a slow (but only modest) improvement. The degree to which they were hyperglycemic initially deteriorated but then improved somewhat by six to eight months (fasted blood glucose of 180–250 mg/dL at eight months; Figure 3). Analysis of the void spot data revealed some striking temporal changes that were highly gender-specific (Figure 6). The males and females handled additional urine volume loads (that were highly correlated with the degree of hyperglycemia—Figure 7) quite differently. The females consistently voided the same number of times regardless of urine volume load, but to adapt, they increased the volume/void (Figure 6C). There was no sign of deteriorating continence (microvoids), and histologically, the bladders looked relatively normal. We conclude that females of this strain represent an example of hyperglycemia-driven polyuria and increased bladder accommodation without overt DVD. The males, in contrast, may be the best of the three models in terms of recapitulating elements of the temporal progression of DVD, usually characterized, in order, by overactivity, underactivity, and loss of outlet control. These results are consistent with the one other study we could find on bladder dysfunction in KK-Ay mice [44]. The authors tested male mice at 12, 18, and 25 weeks of age and noted increased voided volume, frequency of micturition, fasted blood glucose, and glucose intolerance.

### 4.5. Other Observations

A point of note, and potentially one of concern for the assessment of long-term bladder studies in male mice, was the presence of hard white masses in some bladders at sacrifice. TallyHo males did not exhibit these, but they were very obvious in some KK-Ay’s and virtually all of the NoncNZO’s. The sizes varied and, in some cases, were quite large and filled most of the interior lumen of the bladder. A literature search uncovered a report of obstructive uropathy and hydronephrosis occurring in high numbers of male KK-Ay mice, resulting in high mortality rates after four months of age [45]. These plugs appeared to be coagulated seminal vesicle secretions (with no sign of infection) and have been described before in quite a number of laboratory mouse strains [46,47,48]. The authors of the KK-Ay report speculated that the presence of a diabetic phenotype may have been a contributing factor, but a correlation between the development of the seminal plugs and diabetes was not established. The contention that metabolic perturbations were an important contributor, however, is strengthened by the finding in one study that a low-calorie diet reduced outlet obstruction and death by half compared to mice on high-calorie diets [48]. In this study, we noted the mortalities of each strain in Table 2, and it is possible that obstruction was the cause of death for some of the KK-Ay males. Interestingly, although the NoncNZO male mice virtually all had visible and palpable plugs at sacrifice, none died by 12 months of age. Clearly, genetic factors are important for the penetrance and severity of this phenotype.

The complexities of T2D phenotypes in patients and the spectrum of urological symptoms observed over time make the creation of animal models extremely difficult, and as seen here, the role of gender is both highly relevant and often underappreciated. The causes of DVD are debated, but two prevalent lines of thought posit that (1) chronic hyperglycemia-driven inflammation is important, with vulnerable targets in the bladder that include the vasculature, detrusor, urothelium, and neurons [49], and (2) that oxidative stress is responsible for cell and tissue damage [20]. These are likely to be co-existing and interdependent processes. Indeed, a proteomic study on the detrusor and urothelium of TallyHo male mice (21–26 weeks of age) revealed through pathway and network analysis of dysregulated proteins, that diabetes elevated inflammatory responses, oxidative stress, and tissue remodeling [50]. Despite the abundant literature on these systemic processes, there are emerging signs of more local phenomena that target bladder tissues specifically. When methylglyoxal—a reactive by-product of glycolysis found at high levels in diabetic plasma—was administered to healthy mice for four weeks, an overactive bladder phenotype was observed in the complete absence of hyperglycemia [51]. Furthermore, an intriguing recent study from Chen et al. [52] proposed a novel theory on the etiology of DVD. Using a smooth muscle-specific insulin receptor knockout mouse, they showed that voiding phenotypes resembling DVD could be induced despite the mutation having no effect on blood glucose levels. This implied that tissue-specific (i.e., bladder smooth muscle) insulin resistance could induce organ dysfunction locally through insulin’s specific effects on particular cell types in the absence of systemic perturbations to insulin signaling.

This study was not designed to investigate underlying mechanisms in DVD, but from the information obtained, it will now be possible to choose from among several models of naturally occurring T2D with some certainty as to the timing and severity of symptomatic presentation. We would suggest that KK-Ay male mice present the most compelling model for exploring temporal changes in bladder function that resemble aspects of human DVD. The main caveat is that close attention needs to be paid to the potential for obstructive uropathy, and appropriate safeguards built into experimental designs, e.g., avoiding the use of high-fat diets may be helpful. The KK-Ay females, over the course of these experiments, demonstrated polyuric compensation, i.e., larger void volumes. Evidence of later decompensation might be obtained if females were observed for longer than nine months.

## 5. Conclusions

Three type 2 diabetic mouse models were tested for up to 12 months to monitor the appearance of voiding dysfunction. Of the three, KK-Ay male mice developed an overactive bladder phenotype at four months, which then progressed to underactivity by six months and was accompanied by incontinence. As such, these mice most closely resemble the changing symptom profile experienced by human patients with type 2 diabetes. NoncNZO mice did not develop signs of voiding dysfunction even after 12 months. TallyHo females were also completely normal, while males became polyuric at two months and increased their void volumes with no sign of decompensation or incontinence even after nine months. Therefore, KK-Ay males appear to be the best model for investigating the underlying mechanisms of bladder dysfunction in type 2 diabetes.

## Figures and Tables

**Figure 1 metabolites-13-00710-f001:**
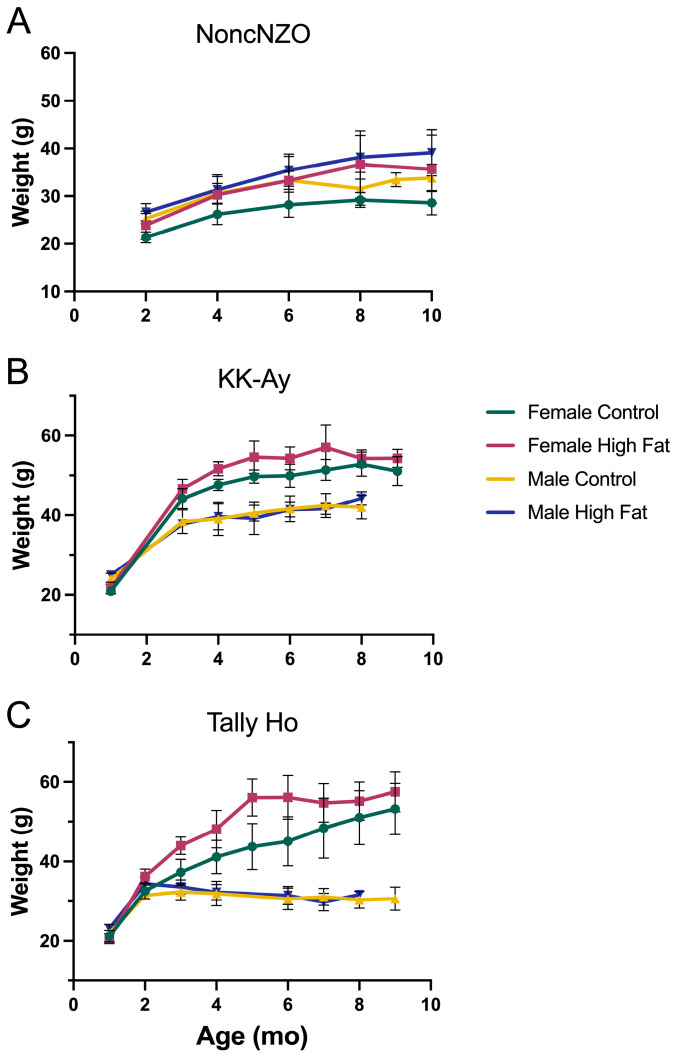
Body weights of diabetic mouse models. (**A**) NoncNZO, (**B**) KK-Ay, (**C**) TallyHo.

**Figure 2 metabolites-13-00710-f002:**
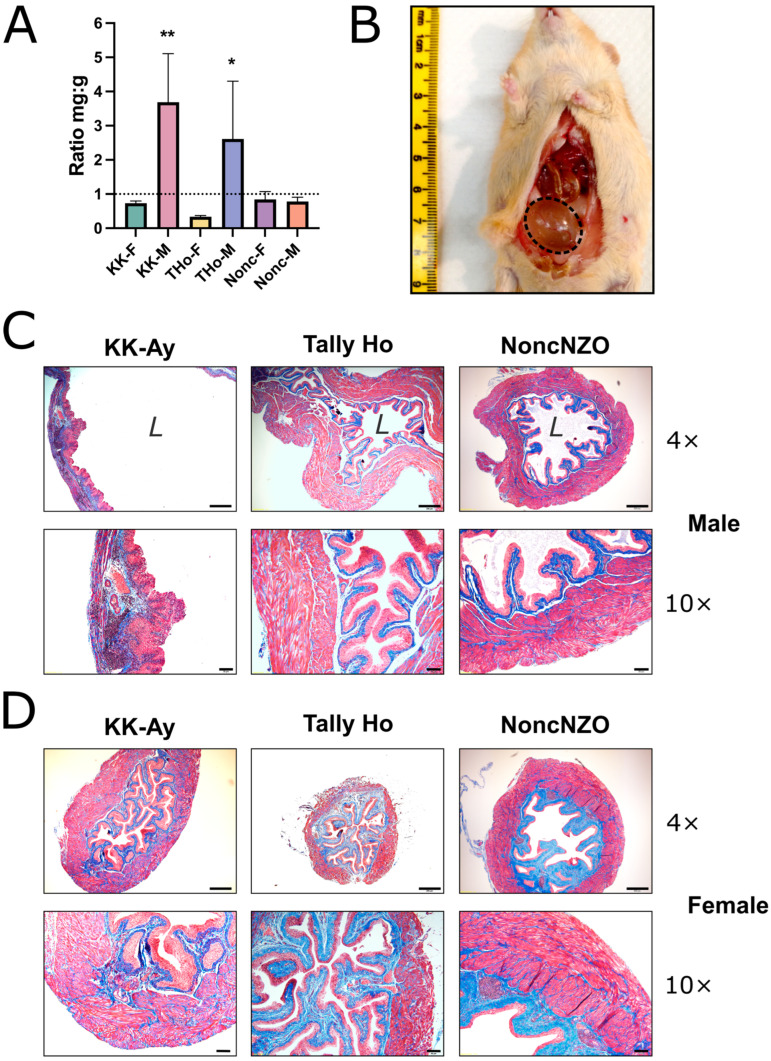
Bladder morphology of diabetic mouse models. (**A**) Bladder:body weight ratios; KK = KK-Ay mice, THo = TallyHo mice, Nonc = NoncNZO mice, F = female, M = male. Dotted line at a ratio of 1.0 shows the result typically found for C57BL/6J control mice. Results for control and high-fat diet mice were combined. Data are mean ± SD (*n* = 5–9), and statistically significant differences between males and females of each strain are shown. * *p* < 0.05, ** *p* < 0.01. (**B**) Example of KK-Ay male mouse bladder—dashed line. (**C**,**D**) Masson’s trichrome staining of mid-bladder paraffin sections from males (**C**) and females (**D**) of all three strains at two magnifications (4× and 10× objectives). *L* = lumen of bladder, scale bars = 200 mm (4×) and 50 mm (10×).

**Figure 3 metabolites-13-00710-f003:**
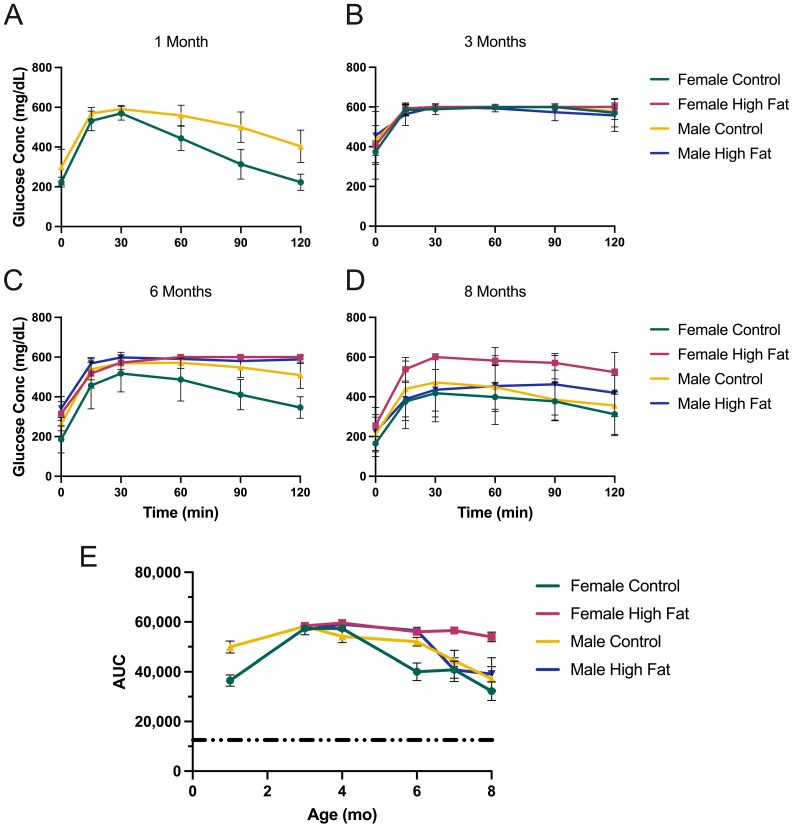
Glucose tolerance testing of KK-Ay mice. (**A**) GTT results at one month of age. As this was the start of the study, mice were on regular chow (*n* = 12 male and 12 female), (**B**) three-month-old mice, (**C**) six-month-old mice, (**D**) eight-month-old mice, (**E**) area under the curve for each group at each time point. Dashed line illustrates the AUC value observed for C57BL/6J male control mice. Data are mean ± SD, *n* = 6 mice/group after one month.

**Figure 4 metabolites-13-00710-f004:**
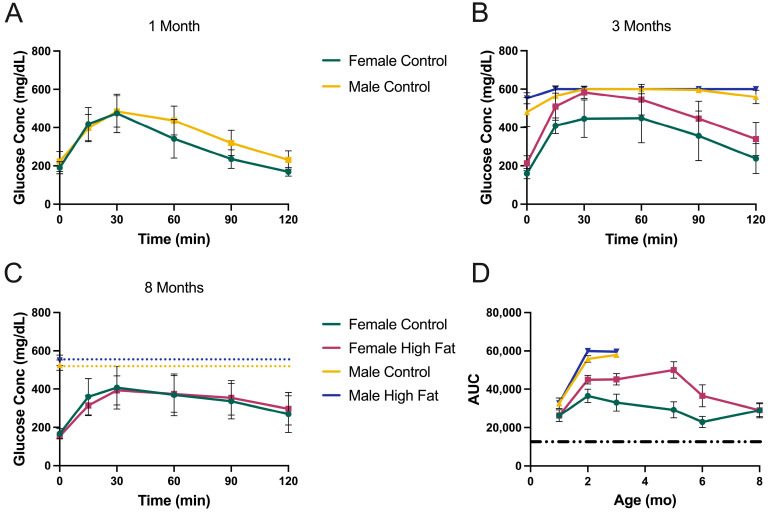
Glucose tolerance testing of TallyHo mice. (**A**) GTT results at one month of age. As this was the start of the study, mice were on regular chow (*n* = 12 male and 12 female), (**B**) three-month-old mice, (**C**) eight-month-old mice. Males were not tested by injection of glucose after three months because their blood glucose levels were already extremely high, as shown by the dotted lines. (**D**) Area under the curve for each group at each time point. Dashed line illustrates the AUC value observed for C57BL/6J male control mice. Data are mean ± SD, *n* = 6 mice/group after one month.

**Figure 5 metabolites-13-00710-f005:**
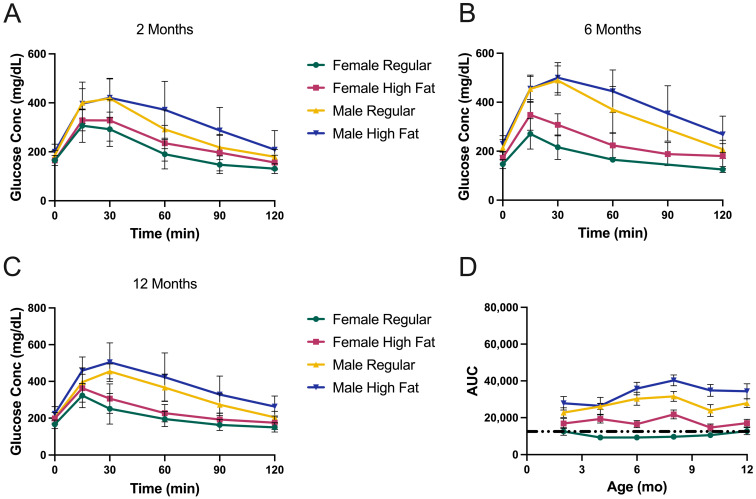
Glucose tolerance testing of NoncNZO mice. (**A**) GTT results at two months of age, (**B**) six months, (**C**) twelve months. (**D**) Area under the curve for each group at each time point. Dashed line illustrates the AUC value observed for C57BL/6J male control mice. Data are mean ± SD, *n* = 10 mice/group.

**Figure 6 metabolites-13-00710-f006:**
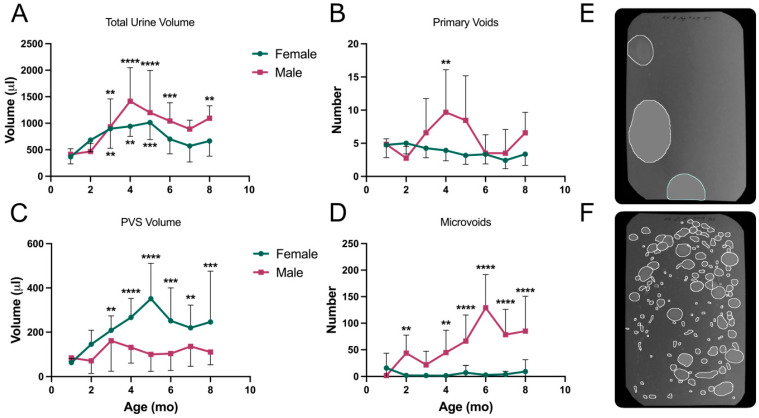
VSA data for KK-Ay mice. Results for control and high-fat diet mice were combined. (**A**) Total urine volume, (**B**) number of primary voids, (**C**) primary void spot volumes, (**D**) number of microvoids, (**E**) representative filter paper image from a five-month-old female, (**F**) representative filter image from a five-month-old male. Filter paper dimensions were 278 × 170 mm (standard mouse cage floor). Data are mean ± SD, *n* = 12/group (7–12 for males due to mortality at later ages). ** *p* < 0.01, *** *p* < 0.001, **** *p* < 0.0001.

**Figure 7 metabolites-13-00710-f007:**
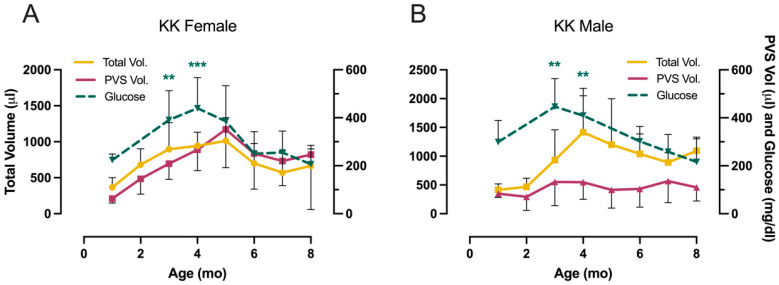
Simultaneous comparisons of total urine volume, primary void spot volume, and fasted blood glucose for KK-Ay mice. (**A**) females, (**B**) males. Data are mean ± SD, *n* = 12/group. Statistical significance is not indicated for total volume and PVS volume, as this can be seen in Figure 6. Means at each age were tested for significance against results at one month. ** *p* < 0.01, *** *p* < 0.001.

**Figure 8 metabolites-13-00710-f008:**
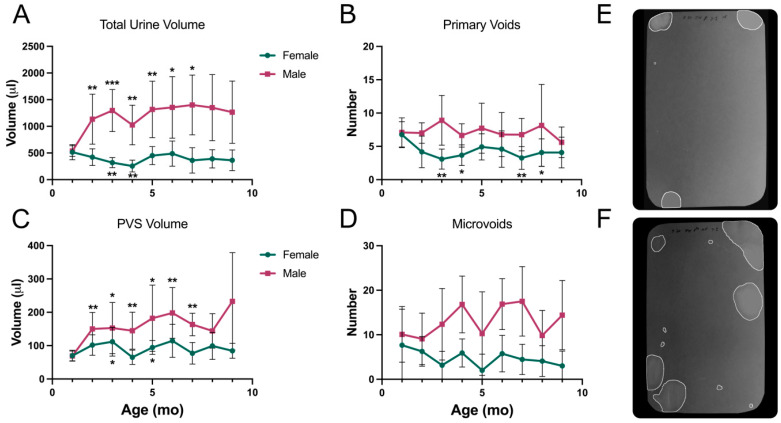
VSA data for TallyHo mice. Results for control and high-fat diet mice were combined. (**A**) Total urine volume, (**B**) number of primary voids, (**C**) primary void spot volumes, (**D**) number of microvoids, (**E**) representative filter paper image from a five-month-old female, (**F**) representative filter image from a five-month-old male. Filter paper dimensions were 278 × 170 mm (standard mouse cage floor). Data are mean ± SD, *n* = 12/group (5–12 for males due to mortality at later ages). * *p* < 0.05, ** *p* < 0.01, *** *p* < 0.001.

**Figure 9 metabolites-13-00710-f009:**
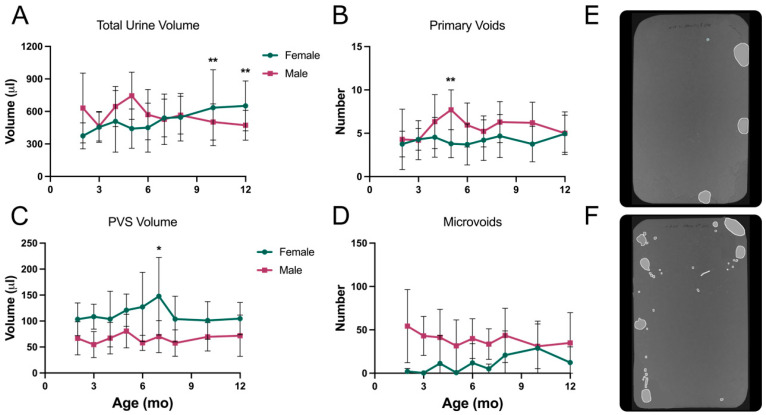
VSA data for NoncNZO mice. Results for control and high-fat diet mice were combined. (**A**) Total urine volume, (**B**) number of primary voids, (**C**) primary void spot volumes, (**D**) number of microvoids, (**E**) representative filter paper image from a five-month-old female, (**F**) representative filter image from a five-month-old male. Filter paper dimensions were 278 × 170 mm (standard mouse cage floor). Data are mean ± SD, *n* = 10–20/group. * *p* < 0.05, ** *p* < 0.01.

**Table 1 metabolites-13-00710-t001:** Characteristics of the three T2D mouse models used.

Strain	Pancreatic Insulin Content	Obesity	Hyperglycemia	Glucose Intolerance	Insulin Resistance
KK-AY	High	Moderate	Moderate but varies with age	Severe	Yes
NONC	Intermediate	No	Moderate	Males—moderate Females—no	Yes
TALLYHO	Low	Females—yes Males—no	Males—severe Females—moderate	Males—severe Female—moderate	Yes

Results based on [34] and observations from this study.

**Table 2 metabolites-13-00710-t002:** Number of deaths at indicated age.

	KK-AY (AT 9 MO)	TALLYHO (AT 9 MO)	NONCNZO (AT 12 MO)
Male control diet	2/6	1/6	0/10
Male high-fat diet	3/6	4/6	0/10
Female control diet	1/6	0/6	4/10
Female high-fat diet	2/6	0/6	6/10

## Data Availability

The data presented in this study are available on request from the corresponding author. Data is not publicly available due to privacy.

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
