# Peer review of "A Spectrum of Age- and Gender-Dependent Lower Urinary Tract Phenotypes in Three Mouse Models of Type 2 Diabetes"

_metabolites, 2023, doi:10.3390/metabo13060710_

Round 1

Reviewer 1 Report

The manuscript metabolites-2316245 by Bryce MacIver et al, entitled “A Spectrum of Age-Dependent Lower Urinary Tract Pheno-types in Mouse Models of Type 2 Diabetes” show that three different polygenic T2D strains, monitored monthly, by void spot assay (VSA) and glucose tolerance testing, to identify which strains would most closely mimic the human disease in terms of an evolving phenotype and to define the time-line and dietary regime necessary for the onset of symptoms.

The manuscript is written in a fluent and correct form and the experiments are well designed and well explained. The Introduction is short but well designed.

The results obtained by authors offer the possibility to choose from among several models of naturally occurring T2D with some certainty as to the timing and severity of symptomatic presentation.

I have just some minor comments:

-       line 131: “… and consistency with previous published data from our group (refs?).”; please, insert the missing reference.

-       Figure 4: please, adjust the legend about the graphs.

-       In many places in the text the microliter is represented by an unknown symbol; please, correct.

-       I didn’t well understand why the authors use different time of age for different strains (at either 4 weeks of age (KK-Ay 158 and TallyHo) or 8 weeks of age (NoncNZO)); please explain.

-       Why not consider including supplementary figure 1 in the manuscript (in figure 2)?

-       Why for the AUC analysis the authors considered the standard error differently to other types of analysis in which they used standard deviation?

After these minor revisions to the manuscript, in my opinion, the manuscript could be accepted in Metabolites journal.

Author Response

  1. Thank you for pointing out the oversight. The appropriate references have now been added at line 131;
  2. Legend 4 has now been edited to clarify;
  3. The unknown symbol was added by the journal, presumably, this occurred during the conversion of the font to Palatino linotype. We will check carefully during final proofreading to make sure the error is not perpetuated;
  4. The different age of one strain (NoncNZO - 8 weeks instead of 4 weeks) occurred because it was the earliest age of availability from the Jackson Lab. This has now been added to Section 2.2 of the Materials and Methods;
  5. We concur with the reviewer about Supplementary figure 1 and have now restructured figure 2 to show all of the histology in one main figure.
  6. The AUC analysis has data presented with standard error simply because Prism software presents the results in that form. This has now been explained in Materials and Methods, section 2.7.

Reviewer 2 Report

The abstract should be a total of about 200 words maximum

Line 35: write in full DM

Line 46: add reference

Line 117: insert in the text the reference to table 1

Line 131: what does mean (refs?)?

Line 139: write the micron and the orientation of the slide

Line 140: add reference of standard method

Line 141: write in full BPS

Line 141: add catalogue number

Line 162: add a figure/scheme showing a timeline of all experiments

Line 206: remove bold A) Bladder:b

Line 254: move figure 4 after line 254

Author Response

Each of the suggestions listed by the reviewer has been implemented in the revised manuscript and may be seen in the tracked changes version of the .doc file.

The one suggestion we wish to decline is to add a schematic figure showing the timeline of experiments. This is because there is nothing complicated in the experimental design.  We tested three strains of mice monthly for void spot assays, body weight and blood glucose and at the end of the study period obtained bladder weights and histology. We believe adding another figure showing timeline would be superfluous.  However, if the editor or reviewer strongly disagrees we could certainly include one.

Reviewer 3 Report

Even though the idea sounds interesting, there is no clear purpose behind the design of this experimental study. What is the main objective? Determine what is the phenotype of type 2 diabetic mice whose symptoms resemble those of patients with DM2 who have urination dysfunction? Why is it important to find this model? What would be the ultimate goal when finding it? What advantages does it offer? How does current knowledge help? What suggestions on diagnosis, treatment, prevention could offer?

 Major points

1.       The manuscript needs writing editing. It would be better to improve the title, it should reflect the content of this experimental study conducted in mouse. What are the authors trying to convey, the importance of bladder dysfunction in patients with DM2/obesity or the need to use mouse models? What is the main aim? Why did the authors characterize three polygenic mouse models of T2DM? The main objective must be the same throughout the manuscript (abstract, introduction and results/discussion). Lines 21-23: Rephrase. Urinary incontinence? Authors should not use the words that appear in the title as keywords. References must be recent and relevant.

2.       The introduction section: Although the authors have described the issues involved in this manuscript, they do not highlight with this information the main objective of this experimental study. Why was it necessary to create these mouse models, to understand what?. What does ER mean? The main objective must be the same throughout the manuscript. Lines 79-83: This information is part of the M&M section. The research question should be clearly outlined. A good and clear justification for conducting this study should be given. It would be better if the authors offered the hypothesis before the main objective of this study. What do the authors hope to find with this experimental study? Lines 85-88: It would be better to use this sentence in the discussion section as a suggestion.

3.       The materials and methods section needs improvement. More details (written in the results section) in each subsection are required to replicate this study. What kind of study was it? What mouse evaluation data was collected and how often? Was there a control group? How long was this study designed for? Line 93: Two types of diet were used. Irradiated... It would be better to use high-fat diet. Lines 13-105: Improve this description. If it happens in all three models, write it in a paragraph at the end of this subsection. Line 129: What results were evaluated? (Ref?)? What blood measurements were evaluated? What would be the expected values/cut points for the measurements made? When would the mice be sacrificed? Immunohistochemical studies, right? Of all the tissue samples for total RNA? What was the purpose of studying "mRNA Expression Analysis by Quantitative RT-PCR (qPCR)". All variables studied must be measured and described in detail. The description of the statistical analysis should be improved.

4.       In the results section: A better summary of the most significant results should be made. In the text, the authors should write the most significant results, and they should avoid repeating the same information in the text if this data appear in the tables, figures, etc. If would be better if all procedures, evaluations (weight assessment), etc. and the purpose for which they were made must be clearly explained/described in the M&M section. Lines 154-162: This description is part of the study design methodology and can be added in the corresponding subsections. Line 165-166: Was this difference significant? Line 170-171: Delete this sentence and use it in the discussion section. The importance of the result should be argued in the discussion section. Lines 171-178: Write it in a separate paragraph. Was mortality significantly higher in groups B and C compared to group A? Lines184-185: Add this information from the M&M section and what assessments were done on the bladder. Lines 185-187: Add this justification to the introduction or discussion section. Line 188: Were these dramatic differences significant? Line 190-191, 226-228, 230-232, 327, 346-347, 348: Write this description in the discussion section. Line 193: What proportion of the mouse body does this bladder size correspond to? The fifth part? What would be normal? Lines 198-199: It would be better if the authors compared males (2c) and females (2d) on the same line. All procedures (design sequence) should be described in the M&M section. Line 208-210, 216-222, 275-281, 286-290, 308-309, 311-314, 334-337: This description must be written in detail in the M&M section. Lines 215 and 272: Give the most significant results by comparing the three groups studied with each other and with the control group. Line 238: If the authors use this mouse model to compare their results, this should be written as part of the study design in the M&M section. Line 225-226: What should be the regular fasting glucose value? Write the normal range in the M&M section. Line 243-244: Enter this glucometer data (maximum reading) in the M&M section. Line 282-284: Why did the authors decide to do this? Line 298: Was it urine volume per day, per void? Line 315-318: Add the need for this analysis in the M&M section. Line 330: Is it seen as leaks?

5.       The discussion needs deep improvements. This section should start with the main objective of this study and the most significant results. The discussion should be more argumentative about the main objective and the most significant results of this study. The results must be discussed from multiple angles and placed in context without being over-interpreted. How are these results comparable to diabetic patients with urination dysfunction, age, severity? Lines 376-378: The needs of all researchers? Shouldn't the primary goal be to try to understand the physiology and ultimately improve a chronic condition that frequently occurs in T2DM patients, through mouse models? Line 382: which might be comparable to those untreated diabetic patients. Line 397: Was it significant? Lines 405-406: Another point to take into account would be to study the characteristics of this mouse model that prevent it from presenting diabetic voiding dysfunction despite not receiving treatment. Line 410-427: What happens with voiding dysfunction? Line 848: Despite the histological studies, the authors did not study whether he had an inflammatory state in the bladders, right?

 It would be a good idea to discuss the results in subsections and use the original titles of the results section here. This is the adequate place where authors can show with arguments the conclusions they reached each step. A paragraph of strengths/limitations and suggestions for this study should be written before the conclusion.

6.       It would be better if the conclusions were written in a different section. The conclusion should improve and should be the same as the abstract.

 I encourage the authors to rewrite the manuscript, thinking about the principal goal of this study, and its design and answering with the results and arguments of the discussion the most proper conclusion to this research work. It should be clear why this study is important and how it improves existing knowledge on the subjects covered.

Author Response

We appreciate the degree of attention paid by the reviewer and the detailed set of critiques. Unfortunately, the main point of the paper appears to have eluded them, perhaps due to unfamiliarity with the state of the field in diabetic bladder dysfunction.  In line with the editors comments we have chosen not to attempt a point by point response.

Round 2

Reviewer 3 Report

Even though some modifications were made, these have not been enough. The study is very interesting, however, to better understand the objective of this experimental study, modifications must be made to the wording of the manuscript. The main problem is that in the results section, the authors write a series of detailed descriptions (highlighted in the text) that are part of the Material and Methods section, for example, the mouse model with which the authors compare their results. Finally, it should be clear from the discussion of the results the conclusion they write in the abstract.

In the abstract: It would be better to write "Therefore, the main objective of this experimental study was to characterize diabetic bladder dysfunction in three promising polygenic mouse models of type 2 diabetes. We performed periodic assessments of glucose tolerance and micturition (void spot assay), for eight to twelve months."

The introduction section: It would be a good idea to rewrite it this way: "We believe that the finding of a mouse model would be very useful to understand the intrinsic mechanism of diabetic bladder dysfunction. Therefore, the main objective of this experimental study was to characterize diabetic bladder dysfunction in three polygenic strains of T2D. The primary endpoint was to identify which strains would most closely resemble the human disease in terms of an evolving phenotype, and the second endpoint was to define the timeline and dietary regimen required for symptom onset."

The material and methods section: To achieve our goal, we used three polygenic strains of T2D, using regular glucose tolerance and VSA tests for 8 to 12 months. The three strains, KK.Cg-Ay/J (KK-Ay), TALLYHO/JngJ (TallyHo) and NONcNZO10/LtJ (NoncNZO) were selected to represent different aetiologies of T2D. Both men and women were evaluated, as well as the effect of normal and high-fat diets. Here, the authors should describe that this specific mouse was used as a model to compare their results.

The results section. There are many descriptions written in the “previous report”, which are part of the design and need to be written in the M&M section to make it completer and more detailed.

The discussion section should improve. This section should start with the main objective of this study and the most significant results.

 It would be better if the conclusions were written in a different section. The conclusion should improve and should be the same as the abstract.
